# Identification of the Potential Role of the Rumen Microbiome in Milk Protein and Fat Synthesis in Dairy Cows Using Metagenomic Sequencing

**DOI:** 10.3390/ani11051247

**Published:** 2021-04-26

**Authors:** Xin Wu, Shuai Huang, Jinfeng Huang, Peng Peng, Yanan Liu, Bo Han, Dongxiao Sun

**Affiliations:** 1National Engineering Laboratory for Animal Breeding, Key Laboratory of Animal Genetics, Breeding and Reproduction of Ministry of Agriculture and Rural Affairs, Department of Animal Genetics, Breeding and Reproduction, College of Animal Science and Technology, China Agricultural University, Beijing 100193, China; wuxin0208@126.com (X.W.); huangjinfeng1003@126.com (J.H.); pengpeng940203@126.com (P.P.); liuyanan961102@126.com (Y.L.); hanbo_98@126.com (B.H.); 2State Key Laboratory of Animal Nutrition, Beijing Engineering Technology Research Center of Raw Milk Quality and Safety Control, College of Animal Science and Technology, China Agricultural University, Beijing 100193, China; huangshuai510@126.com

**Keywords:** dairy cattle, milk protein, milk fat, metagenome sequencing, rumen microbiome

## Abstract

**Simple Summary:**

The rumen is the main digestive and absorption organ of dairy cows. It contains abundant microorganisms and can effectively use human-indigestible plant mass. Therefore, we used metagenomics to explore the role of rumen microbes in the regulation of milk protein and fat in dairy cows. This study showed that *Prevotella* species and *Neocallimastix californiae* in the rumen of cows are related to the synthesis of milk components due to their important functions in carbohydrate, amino acid, pyruvate, insulin, and lipid metabolism and transportation metabolic pathways.

**Abstract:**

The rumen contains abundant microorganisms that aid in the digestion of lignocellulosic feed and are associated with host phenotype traits. Cows with extremely high milk protein and fat percentages (HPF; *n* = 3) and low milk protein and fat percentages (LPF; *n* = 3) were selected from 4000 lactating Holstein cows under the same nutritional and management conditions. We found that the total concentration of volatile fatty acids, acetate, butyrate, and propionate in the rumen fluid was significantly higher in the HPF group than in the LPF group. Moreover, we identified 38 most abundant species displaying differential richness between the two groups, in which *Prevotella* accounted for 68.8% of the species, with the highest abundance in the HPF group. Functional annotation based on the Kyoto Encyclopedia of Gene and Genome (KEGG), evolutionary genealogy of genes: Non-supervised Orthologous Groups (eggNOG), and Carbohydrate-Active enzymes (CAZy) databases showed that the significantly more abundant species in the HPF group are enriched in carbohydrate, amino acid, pyruvate, insulin, and lipid metabolism and transportation. Furthermore, Spearman’s rank correlation analysis revealed that specific microbial taxa (mainly the *Prevotella* species and *Neocallimastix californiae*) are positively correlated with total volatile fatty acids (VFA). Collectively, we found that the HPF group was enriched with several *Prevotella* species related to the total VFA, acetate, and amino acid synthesis. Thereby, these fulfilled the host’s needs for energy, fat, and rumen microbial protein, which can be used for increased biosynthesis of milk fat and milk protein. Our findings provide novel information for elucidation of the regulatory mechanism of the rumen in the formation of milk composition.

## 1. Introduction

Cow’s milk, unmatched by other foods, is full of essential energy elements, such as amino acids and nutrients, ensuring proper human growth and development, including bone formation [1]. With an increase in the world population, milk consumption is expected to rise steadily over the next 50 years [2]. Meanwhile, people have become more aware of milk quality, which is mainly determined by the milk protein and fat content [3,4]. Therefore, intensifying milk production efficiency and quality has been a continued priority [5].

The rumen of dairy cows is the main digestion and absorption organ. Similar to a large anaerobic fermenter, the rumen contains abundant microorganisms, comprising bacteria, protozoa, and fungi, that can use grass, food, and non-protein nitrogen, thereby promoting the growth and development and milk production of animals [6,7,8,9,10]. The microorganisms in the rumen aid in the digestion of complex fibrous substrates into fermentable sugars. Ultimately, the sugars fermented by rumen bacteria are primarily converted into volatile fatty acids (VFA) [11]. Notably, the ruminal VFA and microbial proteins derived from microbial fermentation are the key factors directly affecting milk biosynthesis [12]. Approximately 90% of the rumen VFA are acetate, butyrate, and propionate that are absorbed into the blood through the rumen wall and then transported to the liver. Subsequently, the liver transports cholesterol to the mammary gland for lipid synthesis as lipoproteins [13,14,15,16,17,18,19]. For lactose biosynthesis, glucose produced from the digestion of carbohydrates in the rumen enters the mammary gland through the blood circulation and then is converted into lactose [20]. In a similar way, for milk protein biosynthesis, rumen microorganisms decompose feed and synthesize microbial proteins in the rumen. The microbial proteins and undegraded dietary proteins that are digested into amino acids are transported to the liver and then finally transported to the mammary gland via the blood circulation to synthesize milk protein [21].

Metagenomics is the analysis of microbial communities in particular habitat(s) employing high-throughput sequencing without the obligation of laboratory culture and isolation of individual strains [22,23]. It has been widely used to examine the microbial diversity and metabolic capabilities of microbes in different ecological niches, fermented food, wastewater treatment facilities, and the gastrointestinal tract in humans and animals [20,21,24,25,26,27]. In ruminants, rumen microbes play a vital role in the decomposition of plant lignocellulosic matter [28,29,30]. Xue et al. [31] revealed that rumen bacterial richness and the relative abundance of several bacterial taxa were significantly different between dairy cows with high and low milk protein yields, suggesting the importance of rumen microbiota for milk protein yield. Jami et al. [32] examined the degree of divergence between distinct dairy cows and found that certain physiological parameters, such as milk yield and milk fat yield, highly correlate with the abundance of various bacteria in the rumen microbiome. So far, most studies through 16S rRNA sequencing in dairy cows have mainly focused on phylum and genus levels of the rumen microbiota to study their effects on milk production, feed conversion ratio, and methane emissions. Notably, investigation of the regulatory roles of the rumen microbiota in dairy cattle on milk composition traits is still limited to the species level [33].

In this study, we examined the relationship between the rumen microorganism composition in lactating Holstein cows with extremely high and low milk protein percentages (PP) and fat percentages (FP), ruminal fermentation, and milk quality parameters that may be contributing to high-quality dairy milk production.

## 2. Materials and Methods

### 2.1. Animals and Sample Collection

Six healthy late-lactating Chinese Holstein cows (Table 1) were assigned to the high milk protein and fat percentage (HPF) or the low milk protein and fat percentage (LPF) group from a pool of 4000 lactating Holstein cows (Table 2) in the Sanyuanlvhe Dairy Farming Center (Beijing, China).

All cows in this experiment were second parity. The cows were sacrificed before morning feeding, and the rumen pH was measured immediately using a portable pH meter (Testo 205, Testo AG, Lenzkirch, Germany). To obtain representative samples, rumen contents were collected from four different parts, the cranial sac, the ventral sac, the caudodorsal blind sac, and the caudoventral blind sac, according to the detailed description of the anatomic structure of the bovine rumen [34] within 15 min of slaughter. The rumen contents were filtered through four layers of cheesecloth and immediately frozen in liquid nitrogen to be stored at −80 °C for further analysis. VFA analysis was performed with gas chromatography by using 1 mL of a corresponding rumen fluid sample, as described previously [35]. Meanwhile, ammonia-N analysis was performed using the phenol hypochlorite colorimetric method [36].

### 2.2. DNA Extraction and Metagenome Sequencing

A total of 5 mL of rumen fluid of each sample was lysed with a shaker, and then total genomic DNA was extracted using the QIAamp DNA Stool Mini Kit (Qiagen Ltd., Hilden, Germany). DNA sample concentrations were measured by a UV–VIS NanoDrop 2000c spectrophotometer (NanoDrop Technologies, Wilmington, DE, USA), and DNA integrity was assessed using 1.0% agarose gel electrophoresis (Appendix A). Subsequently, 1.98~2.55 µg of gDNA was used for library preparation. Sequencing libraries were generated using the NEBNext^®^ Ultra™ DNA Library Prep Kit (NEB, Ipswich, MA, USA) following the manufacturer’s recommendations. Index codes were added to attribute sequences to each sample. Briefly, the DNA sample was fragmented by sonication to a size of 350 bp, and then the DNA fragments were end-polished, A-tailed, and ligated with a full-length adaptor for PCR amplification. Finally, the PCR products were purified (AMPure XP system, Beckman, CA, USA), and libraries were analyzed for size distribution by an (Agilent2100 Bioanalyzer, Agilent, Palo Alto, CA, USA) and quantified using real-time PCR. Metagenome sequencing was performed on an Illumina HiSeq sequencing platform (Illumina Inc., San Diego, CA, USA, 150 bp paired-end sequencing).

### 2.3. Quality Control and Assembly of Metagenomic Data

For subsequent analysis, clean data were extracted from the raw data obtained from the Illumina HiSeq sequencing platform using the Readfq program (https://github.com/cjfields/readfq, Version 8 at 11 October 2011, accessed on 18 February 2021). Low-quality reads of <40 bp length, N bases of 10 bp, and overlaps above 15 bp in length were removed. To eliminate data contamination from the host, the clean data were compared against a reference bovine genome UMD3.1.69. Bowtie (http://bowtie-bio.sourceforge.net/bowtie2/index.shtml, Version 2.2.4, accessed on 18 February 2021) was used to filter the reads of host origin; the parameters were as follows: end-to-end, sensitive, I 200, and X 400. The potential reads from the host were filtered out by applying the following parameters [37]: end-to-end, sensitive, I 200, and X 400. Lastly, the filtered reads were assembled using SOAPdenovo software (http://soap.genomics.org.cn/soapdenovo.html, Version 2.04, accessed on 18 February 2021). Scaffolds of >500 bp were extracted for subsequent genetic prediction and annotation [38,39,40].

### 2.4. Gene and Taxonomy Prediction

For each sample, open reading frame (ORF) prediction in the Scaftigs (≥500 bp) was performed using MetaGeneMark (http://topaz.gatech.edu/GeneMark, Version 2.10, accessed on 18 February 2021) software [41,42,43]. Data redundancy was removed using CD-HIT software (http://www.bioinformatics.org/cd-hit, Version V4.5.8 at September 2009, accessed on 18 February 2021), and a non-redundant initial gene catalog was obtained [44,45]. For taxonomic analysis, DIAMOND [46] (https://github.com/bbuchfink/diamond, Version 0.9.9, accessed on 18 February 2021) was used to blast the unigenes against the bacteria, fungi, archaea, and virus sequences in the Non-Redundant Protein Sequence Database (NR database) of the NCBI (https://www.ncbi.nlm.nih.gov/ at 2 January 2018, accessed on 18 February 2021) using the parameter (blastp, evalue ≤10^−5^).

Using the R package, we used the log2 logarithmic transformation of the species-level abundance value as the ordinate to draw the rank abundance curve of each sample, presenting diversity of the bacterial communities. 

### 2.5. Comparative Analysis of Microorganism Abundance in the HPF and LPF Groups

Krona was used to display the relative abundance and the abundance cluster heat map. Principal component analysis (PCA) [47] (R ade4 package, version 2.15.3) and NMDS [48] (R vegan package, version 2.15.3) decrease-dimension analysis were carried out based on the abundance of the taxonomical hierarchy at genera and species levels. ANOSIM was used to examine the differences in microorganism abundance within and between groups (R vegan package, version 2.15.3 at 22 June 2012). Linear discriminant analysis (LDA) effect size (LEfSe) and Metastats analyses were performed to identify the species with different abundance between groups at the species level. LEfSe software (http://huttenhower.sph.harvard.edu/galaxy/, accessed on 18 February 2021) was used with the default threshold LDA score of 2 [49]. In Metastats analysis, for each taxonomy, the permutation test was performed between groups to obtain the raw *p*-Value, which was corrected by the Benjamini and Hochberg false discovery rate to acquire the *q*-Value.

### 2.6. Functional Annotation 

To analyze the functions of rumen microorganisms, the unigenes were blasted using DIAMOND software against functional databases, including the Kyoto Encyclopedia of Gene and Genome (KEGG, http://www.kegg.jp/kegg at 1 January 2018, accessed on 18 February 2021), evolutionary genealogy of genes: Non-supervised Orthologous Groups (eggNOG, http://eggnogdb.embl.de/#/app/home, Version 4.5, accessed on 18 February 2021), and Carbohydrate-Active enzymes (CAZy, http://www.cazy.org at 4 July 2015, accessed on 18 February 2021). The best BLAST hits were subjected to subsequent analysis. LEfSe and Metastats analyses were performed to identify the distinct functions between HPF and LPF groups.

### 2.7. Correlation Analysis

Using the R package, Spearman’s rank correlation analysis was used to calculate the correlation coefficients among the abundances of different ruminal microorganisms between groups, VFA concentrations, and milk components, as well as to detect the correlation coefficient between each rumen liquid sample, based on the gene abundances, with *p* < 0.05 being considered significant.

## 3. Results

### 3.1. Ruminal pH and VFA Concentrations

Rumen pH and NH_3_-N (mg/dL) were similar between the HPF and LPF groups (*p* > 0.05) (Table 3).

The proportions of acetate, propionate, and butyrate, and total VFA concentrations were markedly higher in the HPF group than in the LPF group (*p* < 0.05). However, the proportions of isobutyrate, valerate, and isovalerate were not significantly different between the two groups (*p* > 0.05).

### 3.2. Sequencing of the Rumen Microbiota

A total of 76 Gbp clean data were generated after quality control (Appendix A). After assembling the samples, a total of 1,524,250,289 bp Scaftigs were obtained. After prediction and redundancy removal by MetaGeneMark and CD-HIT programs, we identified a total of 1,078,009 ORFs covering a total length of 667.24 Mbp in scaffolds longer than 500 bp. From these, a total of 278,274 complete genes, accounting for 25.81%, were annotated (Appendix A). The correlation coefficients (R^2^) among the three samples within each group were 0.80 and 0.79, respectively, and R^2^ between HPF and LPF groups was 0.01, indicating the high similarity of biological replicates and divergence between comparison groups (Appendix A).

### 3.3. Taxonomic Composition of the Rumen Microbiota

We obtained the annotated phylum and genus accounting for 80.48% and 66.82% of the unigenes, respectively. Among these, 6977 microorganisms were detected, including 6259 bacteria, 397 eukaryota, 140 archaea, and 181 viruses. The predominant taxonomic compositions of the rumen samples from six cows are displayed at phylum, class, order, family, genus, and species levels in Appendix A. At the phylum level, the dominant bacterial phyla were *Bacteroidetes* (51.4%), *Firmicutes* (8.72%), *Proteobacteria* (5.77%), and *Fibrobacteres* (3.08%); at the genus level, the most abundant bacterial genera were *Prevotella* (38.48%), *Fibrobacter* (3.08%), and *Bacteroides* (2.47%) and the dominant bacterial species were *Prevotella ruminicola* (3.85%), *Prevotella* sp. *Ne3005* (3.33%), *Prevotella* sp. *tc2-28* (2.77%), and *Prevotella* sp. *tf2-5* (2.31%). 

### 3.4. Differential Abundance of the Rumen Microbiome between the HPF and LPF Groups with PP and FP

We found that the species diversity in the LPF group was higher than that in the HPF group, shown by the rank abundance curve (Appendix A). Using PCA and based on the rumen microbiome abundance at genus and species levels, the six cows with extremely high and low milk PP and FP were separated into two clusters (Figure 1A,B).

Principal coordinate 1 accounted for 60.68% and 55.31%, while principal coordinate 2 accounted for 12.81% and 14.21% of the total variation in the two groups, respectively. Furthermore, ANOSIM revealed significant differences in the abundances of rumen microorganisms between the cows of the HPF and LPF groups (R = 0.889, *p* = 0.1; R = 1, *p* = 0.1; Figure 2).

Among the top 10 most abundant genera, *Prevotella* exhibited the highest proportion in the HPF and LPF groups, at 42.8% and 34.2%, respectively; *Fibrobacter* accounted for 3.13% and 2.69%, and *Bacteroides* accounted for 3.02% and 2.25% in the two groups, respectively. The abundances of another seven genera were 0.44–0.8% (Appendix A). LEfSe analysis detected 289 differential microorganisms (LDA > 2; Appendix A). To find microbiota with more significant differences, we used an LDA value greater than 3 for screening and identified 40 kinds of microbiota with significant differential abundances between the HPF and LPF groups (Figure 3).

Meanwhile, Metastats analysis revealed that 2797 microorganisms showed significant differential abundances between the two groups (*p* = 1.48 × 10^−5^~0.049; *q* = 0.037~0.048). Of these, 38 species were commonly identified by both methods (Figure 4 and Appendix A), where 8 *Prevotella* sp., 2 unclassified_*Bacteroidales*, 2 unclassified_*Bacteria*, and 1 *eukaryota Neocallimastix californiae* exhibited significantly higher abundance in the HPF group, while 25 species were significantly enriched in the LPF group.

Of these, the 35 most abundant microorganisms (abundance > 0.02%; Figure 5) included 16 species of the phylum *Bacteroidetes* in the HPF group and 19 bacteria belonging to four different phyla, namely *Bacteroidetes*, *Firmicutes*, *Fibrobacteres*, and *Proteobacteria*, in the LPF group. Notably, 11 of the 16 species in the HPF group belonged to *Prevotella*, which is involved in carbohydrate degradation.

### 3.5. Correlation among Ruminal Microbiome, Ruminal Fermentation, and Milk Components

Spearman’s correlation analysis revealed the relationship among 38 differentially expressed ruminal microbes, milk components, and the concentrations of the total VFA, acetate, propionate, and butyrate. As shown in Figure 6, we found that *Prevotella* sp. *tc2-28*, *Prevotella* sp. *ne3005*, and *Neocallimastix californiae* were positively correlated with total VFA and acetate (R > 0.5; *p* < 0.01); *Prevotella ruminicola* was positively correlated with PP (R > 0.5; *p* < 0.01); and *Tolumonas lignilytica* and *Succinatimonas sp CAG 777* were negatively correlated with total VFA and acetate (R < −0.5; *p* < 0.01).

### 3.6. Functional Enrichment of the Rumen Microbiome Exhibiting Differential Abundances between the HPF and LPF Groups with PP and FP

To further understand the potential correlation of the rumen microbiome with milk protein and fat traits, the differentially abundant rumen microbiome was functionally annotated using the KEGG, eggNOG, and CAZy programs. Regarding KEGG, LEfSe and Metastats analyses found 66 pathways significantly enriched in third-level pathways (LDA > 2.0, *p* < 0.05; Figure 7 and Appendix A): 48 pathways involved species with a significantly high abundance in the HPF group (4 Metabolism, 8 Cellular Processes, 13 Environmental Information Processing, 14 Organismal Systems, and 9 Human Diseases pathways) while 18 pathways were enriched in the LPF group.

In KEGG orthology analysis, LEfSe and Metastats analyses found 37 significant KOs (LDA > 2, *p* < 0.05; Figure 8 and Appendix A), including 7 KOs (K01190, K05349, K01811, K01847, K03737, K01006, and K00688) that are related to carbohydrate, galactose, starch and sucrose, lipid, sphingolipid, glycan, glutathione, butanoate, pyruvate, energy, and amino acid (AA) metabolism; glycolysis; valine, leucine, and isoleucine degradation; alanine, aspartate, and glutamate metabolism; and the citrate cycle. All of these were enriched in the HPF group, while another 15 KOs that were highly abundant in the LPF group were related to DNA-damage-inducible protein J (K07473), ATP-dependent DNA helicase RecG (K03655), and flagellin (K02406).

Likewise for the eggNOG database, LEfSe and Metastats analyses revealed that eight pathways significantly enriched in the HPF group were involved in carbohydrate transport and metabolism; signal transduction metabolism; energy production and conversion; translation ribosomal structure and biogenesis; nuclear structure; lipid transport and metabolism; secondary metabolite biosynthesis, transport, and catabolism; and posttranslational modification. On the contrary, five significantly enriched pathways in the LPF group were related to replication recombination and repair, amino acid transport and metabolism, coenzyme transport and metabolism, extracellular structures, and cell motility (LDA > 2, *p* < 0.05; Figure 9 and Appendix A).

The CAZy database also revealed a total of 215 CAZyme encoding genes (Appendix A), which included 96 glycoside hydrolases (GHs), 46 carbohydrate-binding modules (CBMs), 45 glycosyltransferases (GTs), 12 carbohydrate esterases (CEs), 11 polysaccharide lyases (PLs), and 5 auxiliary activities (AAs). LEfSe analysis detected 79 CAZyme encoding genes (LDA > 2; Appendix A). Of them, the GH51, GH97, GH31, GH2, GH3, and GH43 family of glycoside hydrolases and CE1 were more significantly enriched in the HPF group, while 12 were enriched in the LPF group (5GH, 1CE, 3GT, and 3CBMs) (LDA > 3, *p* < 0.05; Figure 10 and Appendix A).

Based on the results, the potential regulatory mechanism of rumen microorganisms on milk protein and fat synthesis is shown in Figure 11. In summary, more abundant *Prevotella ruminicola*, *Prevotella* sp. *Ne3005*, *Prevotella* sp. *tc2-28*, and *Neocallimastix californiae* in the HPF group might effectively degrade and ferment the cellulose, hemicellulose, and starch in the rumen better to produce a large amount of pyruvate, which is subsequently transformed into various kinds of short-chain fatty acids (acetic acid, propionic acid, butyric acid, etc.) and amino acids. The metabolin is transferred to the mammary gland to synthesize milk protein and fat through liver synthesis, intestinal absorption, and blood circulation.

## 4. Discussion

In this study, using metagenomics, we investigated the rumen microbiome of Holstein cows in the late lactation stage with extremely high and low milk protein and fat percentages. We found that the differential abundances of the microorganisms at phylum, genus, and species levels were significantly linked to milk protein and fat traits.

We detected a total of 6977 unique microorganisms. The three most abundant phyla were Bacteroidetes, Firmicutes, and Proteobacteria. Notably, these results are consistent with previous discoveries [50,51,52]. Besides, we also found that *Neocallimastigomycota* is the major anaerobic fungus, which is similar to the report by Zhang et al. [51]. On the contrary, based on 16S rRNA sequencing, Huang et al. suggested that Actinobacteria is the most predominant phylum, besides Bacteroidetes and Firmicutes, in the rumen of lactating Chinese Holstein cows [53]. This difference between the studies could be due to the distinct feeding management and environment of the dairy farm where the samples were collected. Consistent with the previous report by Xue et al. [52], we also found that *Prevotella* is the most abundant genus in the rumen of lactating cows with high milk protein.

LEfSe and Metastats analyses revealed that *Prevotella ruminicola*, *Prevotella* sp. *Ne3005*, and *Prevotella* sp. *tc2-28* (LDA > 5, *p* < 0.05) are highly enriched in the HPF group. Notably, *Prevotella* sp. *ne3005* and *Prevotella* sp. *tc2-28* were positively correlated with total VFA and acetate (*p* < 0.01), and *Prevotella ruminicola* was positively correlated with PP (*p* < 0.01). The genus *Prevotella* is known to aid the metabolism of cellulose and starch and the production of acetate, butyrate, and propionate [54,55,56,57,58]. Chiquette et al. [59] reported that *Prevotella bryantii 25A* can improve ruminal fermentation products and milk fat concentration in cows in the early lactation stage. Likewise, Osborne et al. [60] suggested that *Prevotella ruminicola*, along with other cellulolytic bacteria, can synergistically participate in plant cell wall degradation to fully use forage cellulose, hemicellulose, and pectin for conversion into VFA. Overall, these findings suggest that *Prevotella* species are essential for VFA biosynthesis. In our study, a higher concentration of VFA (total VFA, acetate, propionate, and butyrate) in the HPF group than in the LPF group further suggests higher ruminal fermentation efficiency in the former group. In addition, among the 35 most abundant microorganisms, 14 species with the highest abundance in the HPF group belonged to two genera, and 11 of the 14 species belonged to the genus *Prevotella*. In contrast, in the LPF group, 19 species of high abundance were distributed across four different genera. In the HPF group, cows with high milk fat and protein contents showed the characteristics of lower diversity and higher abundance at the genus level. Notably, microbial diversity is an important factor affecting cow metabolism. Shabat et al. [61] found that lower richness of the microbiome gene content and taxa is tightly linked to higher feed efficiency. Interestingly, at the eukaryote level, *Neocallimastix californiae*, which was higher in the HPF group, was positively correlated with the total VFA and acetate. This eukaryote can ease the inner tension of plant fiber, facilitating degradation by rumen microbes [62]. The diversity and abundance of microbes and mutualism between the dominant bacteria and anaerobic fungi create an excellent rumen microbial environment enabling the HPF group to produce more VFA and use sufficient raw materials for milk production.

Based on KEGG and eggNOG databases, we found that galactose, starch and sucrose, glycolysis, lipid, energy, butanoate, and pyruvate metabolism were enriched in the HPF group, indicating the increased production of hydrolytic products and pyruvate due to the improved carbohydrate degradation ability of the microbiome in this group. Compared with the LPF group, the highly abundant genes encoding CAZymes involved in carbohydrate synthesis (GHs) and the higher concentrations of major VFA in the HPF group suggest that their rumen microbiomes are more efficient in hydrolysis to produce VFA and, in turn, improve lactogenesis. Altogether, *Prevotella* and VFA contents in the HPF group were markedly higher than those in the LPF group. The differential *Prevotella* content can alter the carbohydrate, energy, lipid, and amino acid metabolism, causing an increase in acetate, butyrate, and propionate in the HPF group. Acetate and butyrate are milk fat precursors, while propionate is the precursor of milk protein [63,64]. The HPF group microbiome can effectively digest the feed ingredients to degrade them into pyruvate, finally generating more VFA. Acetate and butyrate in the rumen are converted into cholesterol by the liver to later participate in the synthesis of milk fat [13]. Propionate, the main precursor of glucose synthesis in ruminants, is required for energy supply and protein synthesis. An increase in propionate content can also stimulate insulin secretion, blood flow to the breast, and synthesis of milk protein [65,66,67]. Interestingly, we found that pathways related to valine, leucine, and isoleucine degradation and alanine, aspartate, and glutamate metabolism are enriched in the HPF group. Essential AAs are critical for multiple physiological processes [68]. Branched-chain AA (BCAA) supplementation has been found to be beneficial for body weight, lipogenesis, and insulin resistance in several species [69,70,71]. BCAAs are also known to improve milk and body protein synthesis and get oxidized by the tricarboxylic acid cycle to produce ATP during catabolic states [72]. In the HPF group, the enrichment of the tricarboxylic acid cycle suggests an increase in the mutual conversion of body fat, sugar, and protein and the increased synthesis of microbial protein. This may provide sufficient protein to the mammary gland for milk protein synthesis. Future studies must be conducted to identify the functions of differential microbiomes, and single-bacterial cultures should validate the function of *Prevotella* species in regulating milk fat and milk protein. In subsequent experiments, we also plan to test bacteria and eukaryotes with significant differences in enzyme preparations as cow feed to examine their effect on milk traits.

The present study had certain limitations. Only three biological replicates were used for each condition, due to the limited availability of rumen liquid samples from lactating cows, especially the high-production ones. Two previous studies evaluated the relationship between detection capacity and the number of replicates. They found that the true positives, the detection rate (recall) of differentially expressed genes and transcripts, and precision were similar for two or more replicates if most commonly adopted software reads were employed [73,74]. However, more biological replicates should be preferred to achieve broader examinations for improved detection. In addition, the potential regulatory roles of the differentially expressed microorganisms (mainly *Prevotella* species and *Neocallimastix californiae*) need to be further validated.

## 5. Conclusions

In the present study, using microgenomics, we identified the top 38 differentially abundant species between the dairy cows with extremely high and low milk PP and FP, which were involved in carbohydrate, amino acid, pyruvate, insulin, and lipid metabolism and transportation. *Prevotella ruminicola*, *Prevotella* sp. *tc2-28*, and *Neocallimastix californiae*, with higher abundance in the HPF group, were correlated with total VFA and acetate, implying their better capability of digesting feed and providing an adequate substrate for milk synthesis in the mammary glands.

## Figures and Tables

**Figure 1 animals-11-01247-f001:**
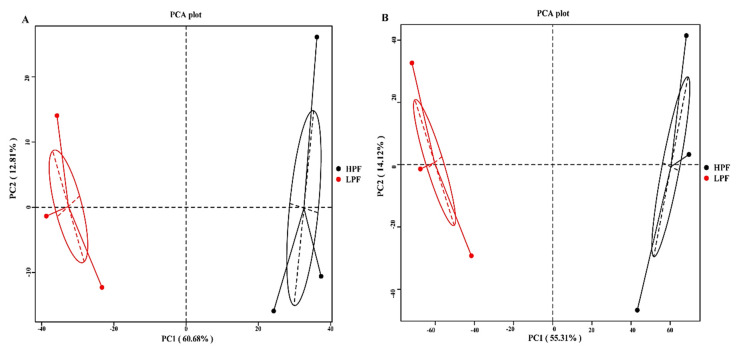
The microbial composition at genus and species levels differs between rumens of high and low milk protein percentages (HPF) and fat percentages (LPF). (**A**) Principal component analysis (PCA) of metagenomics-based genera. (**B**) PCA of metagenomics-based species.

**Figure 2 animals-11-01247-f002:**
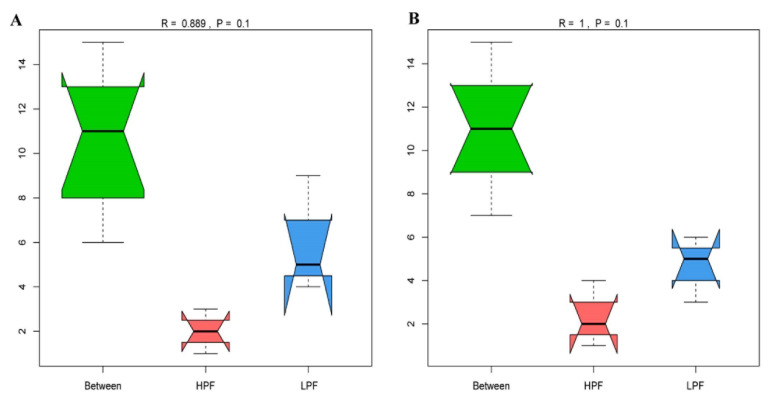
ANOSIM based on genus and species levels. (**A**) ANOSIM based on the genus level. (**B**) ANOSIM based on the species level. The R-value is between (−1,1), and the R-Value is greater than 0, indicating that the inter-group difference was greater than the intra-group difference.

**Figure 3 animals-11-01247-f003:**
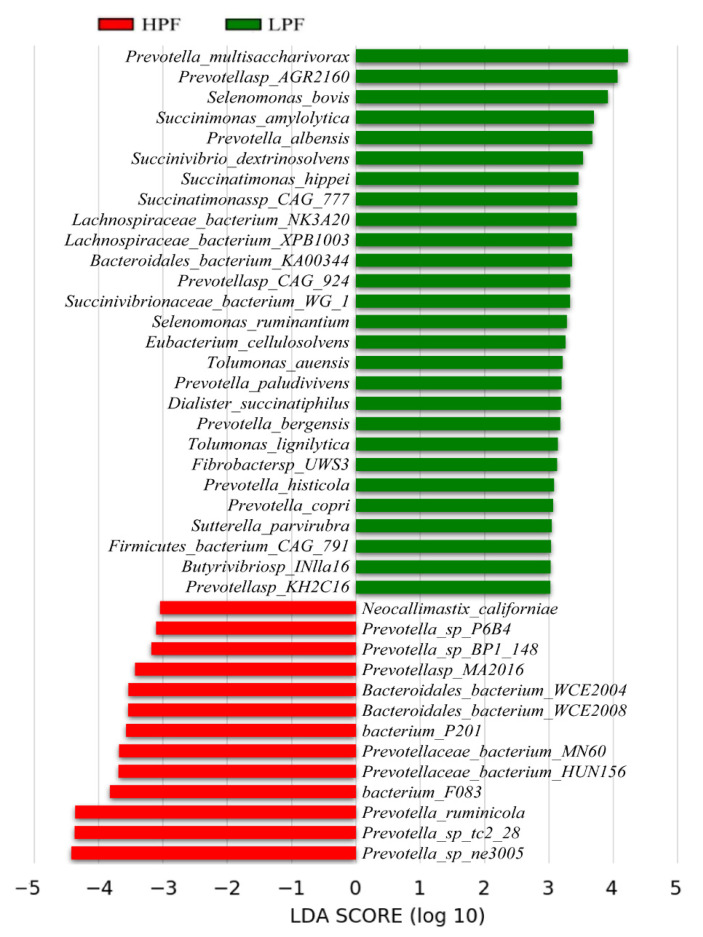
Linear discriminant analysis (LDA) effect size (LEfSe) analysis of rumen microbiota between two groups. Note: An LDA plot indicates the biomarkers found by ranking according to the effect size (3.0) of the species.

**Figure 4 animals-11-01247-f004:**
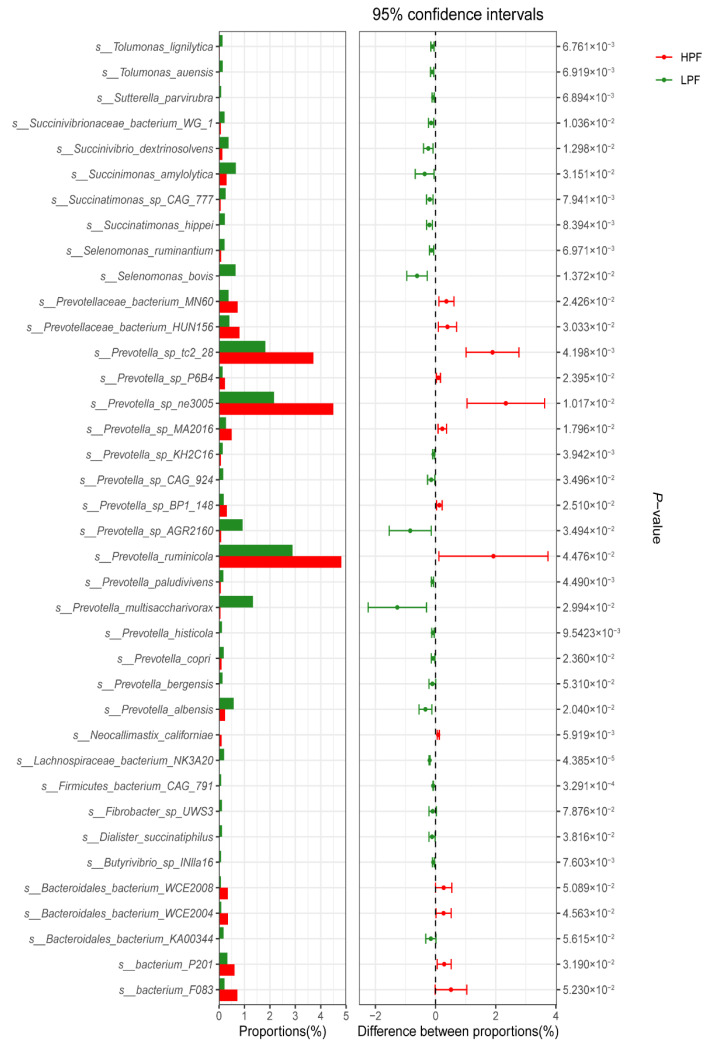
Percentage composition and significance of species in the rumen fluid. HPF, high milk protein and fat percentage; LPF, low milk protein and fat percentage.

**Figure 5 animals-11-01247-f005:**
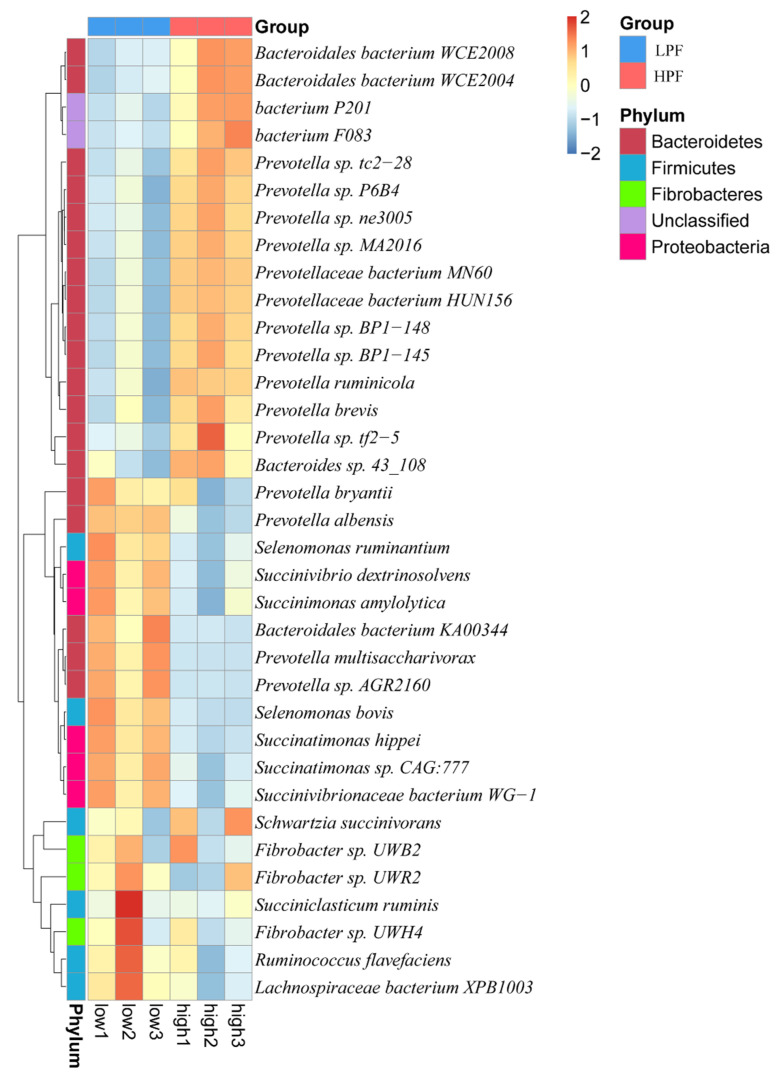
Species-level relative abundance clustering heat map.

**Figure 6 animals-11-01247-f006:**
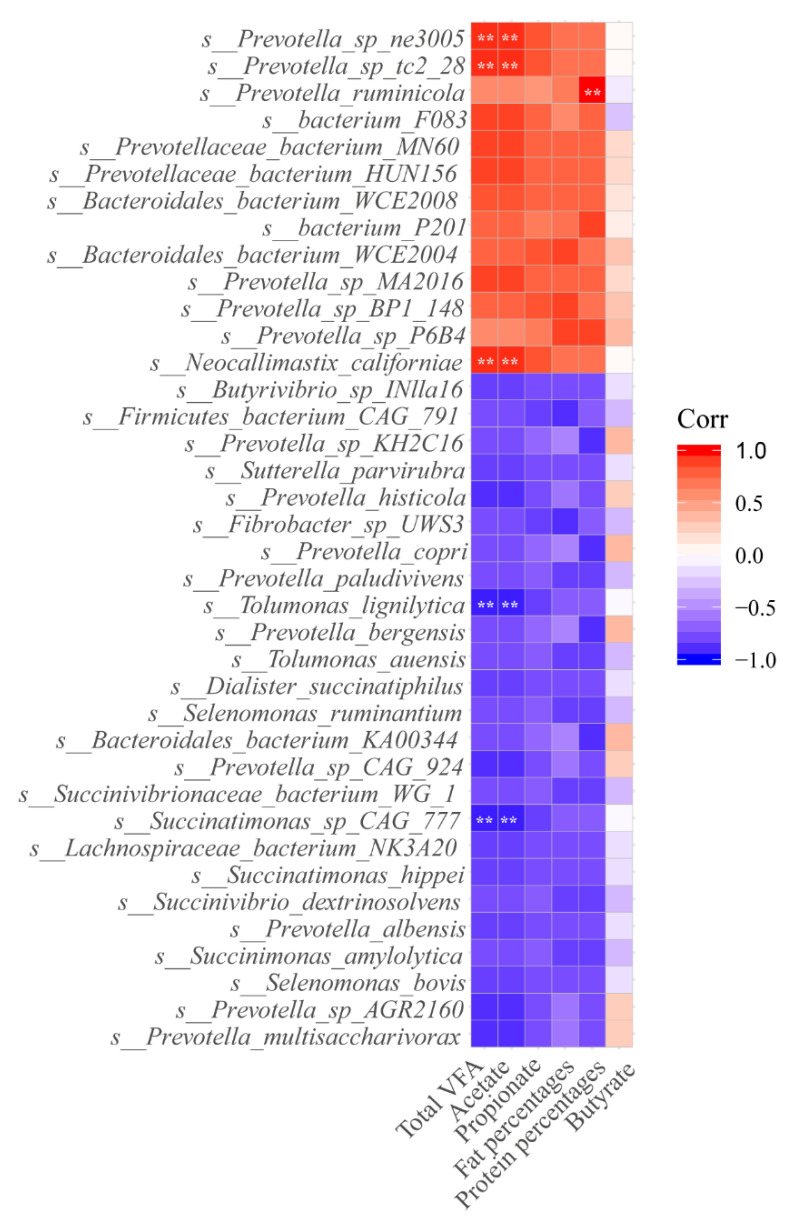
Spearman correlations between specific bacterial species, volatile fatty acids (VFA), and milking traits. Significant correlations: ** *p* < 0.01.

**Figure 7 animals-11-01247-f007:**
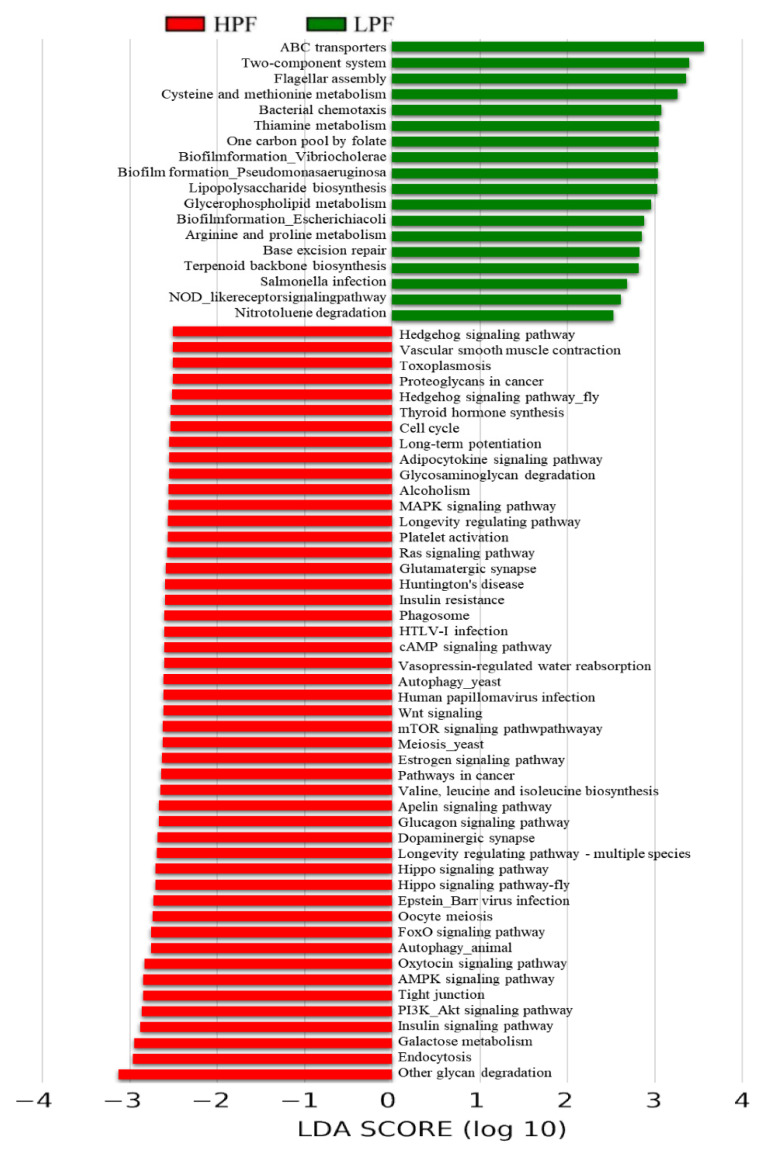
LDA effect size (LEfSe) analysis of the function of the Kyoto Encyclopedia of Gene and Genome (KEGG) between two groups.

**Figure 8 animals-11-01247-f008:**
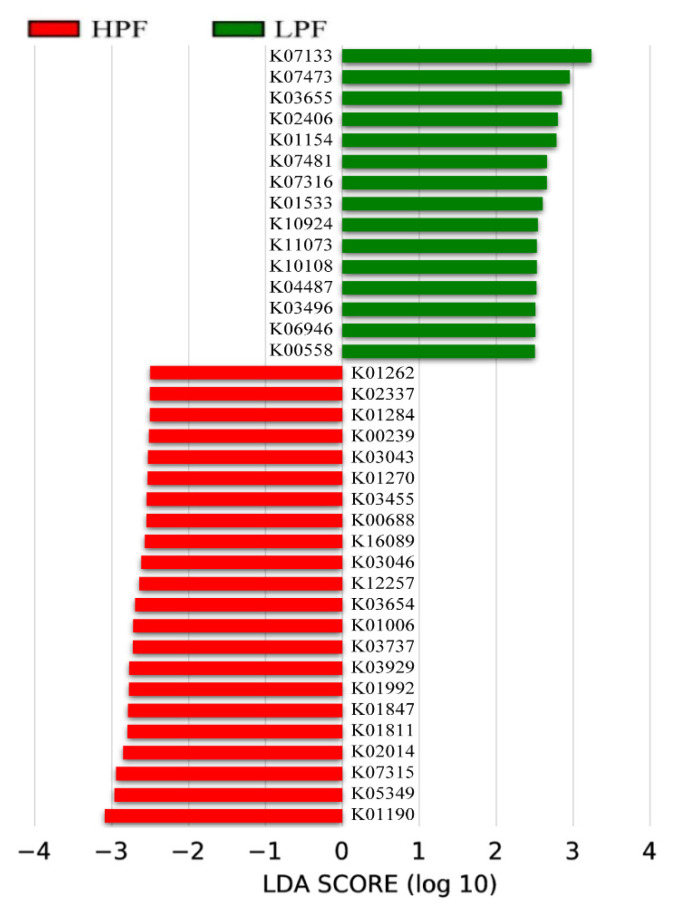
LDA effect size (LEfSe) analysis of the function of KOs between the two groups.

**Figure 9 animals-11-01247-f009:**
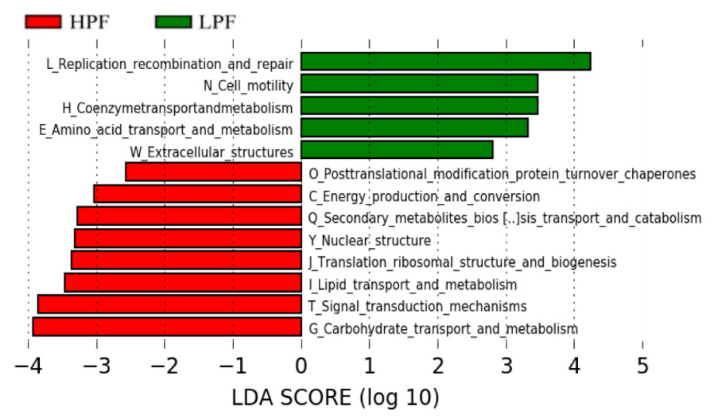
LDA effect size (LEfSe) analysis of the function of evolutionary genealogy of genes: Non-supervised Orthologous Groups (eggNOG) between the two groups.

**Figure 10 animals-11-01247-f010:**
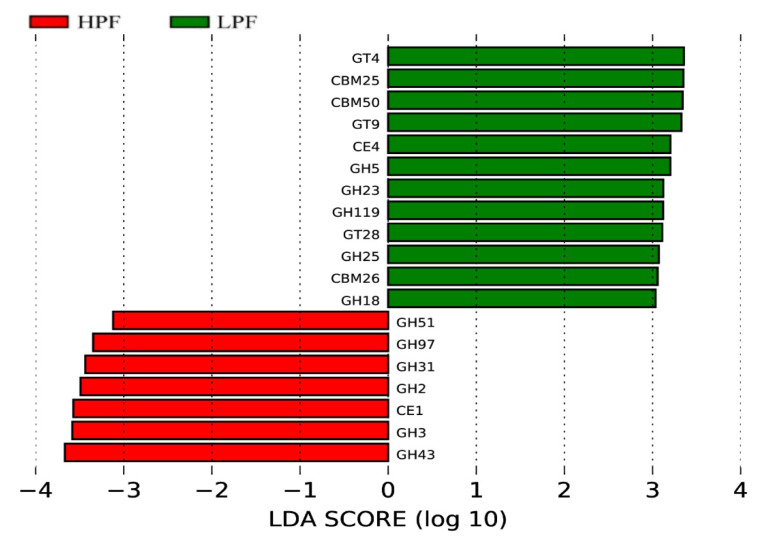
LDA effect size (LEfSe) analysis of the function of Carbohydrate-Active enzymes (CAZy) between the two groups.

**Figure 11 animals-11-01247-f011:**
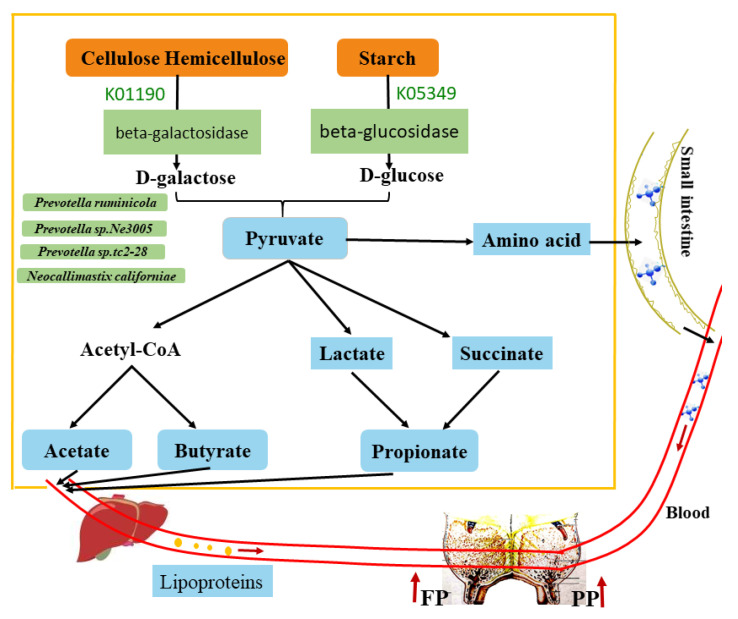
Consolidation of results from the KEGG, CAZy, and eggNOG database analyses. Green: enriched in microbiomes, enzymes, and pathways. Blue: decomposition products. Yellow: plant biomass.

**Table 1 animals-11-01247-t001:** Milk yield, composition, and body weight of dairy cows.

Items	HPF	LPF	SEM	*p*-Value
Milk, kg/d	35.172	34.826	0.851	0.876
DIM, days	244.231	237.336	11.653	0.812
Body weight, kg	637.752	617.210	8.258	0.316
Composition, %				
Fat	4.393	1.940	0.252	0.004 **
Protein	3.943	2.923	0.102	0.002 **

** *p* < 0. 01. HPF, high milk protein and fat percentage; LPF, low milk protein and fat percentage; SEM, standard error of the mean.

**Table 2 animals-11-01247-t002:** Phenotypic values for milk protein and fat percentage of the 4000 Holstein cows.

Groups	Protein Percentage	Fat Percentage
High1	4.02%	4.66%
High2	3.86%	4.68%
High3	3.95%	3.84%
Average of 4000 cows	3.60%	3.10%
Low1	2.78%	1.92%
Low2	3.04%	1.86%
Low3	2.95%	2.04%

**Table 3 animals-11-01247-t003:** Effects of differences between the HPF group and LPF group on metabolites in the rumen.

Items	HPF	LPF	SEM	*p*-Value
pH	6.466	6.606	0.083	0.458
NH3-N, mg/dL	17.956	17.701	0.102	0.221
Proportion	-	-	-	-
Acetate	67.104	47.508	4.599	0.003 **
Propionate	24.703	12.354	2.894	0.003 **
Butyrate	6.247	5.244	0.266	0.035 *
Isobutyrate	0.703	0.519	0.072	0.235
Valerate	0.936	0.809	0.126	0.667
Isovalerate	0.378	0.345	0.019	0.447
Total VFA, mmol/L	100.071	66.780	7.754	0.002 **

* *p* < 0.05, ** *p* < 0.01. SEM, standard error of the mean.

## Data Availability

The rumen metagenome sequences were deposited at the NCBI Sequence Read Archive (SRA) under accession number PRJNA656673 (https://www.ncbi.nlm.nih.gov/bioproject/PRJNA656673, accessed on 18 February 2021).

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
