# Peer review of "Identification of the Potential Role of the Rumen Microbiome in Milk Protein and Fat Synthesis in Dairy Cows Using Metagenomic Sequencing"

_animals, 2021, doi:10.3390/ani11051247_

Round 1
Reviewer 1 Report
Overall, this article is organized in a favorable manor, provides relevant data that has implications in production setting, and seems to have support for most conclusions drawn. I have gone through each section and identified some areas for potential improvement.
Abstract:
Line 26: "ruminal contents of volatile fatty acids..." this sentence could use clarification and revisions. Also, perhaps refer to total concentration of volatile fatty acid if including in a list with the specific VFA found to be increased in production.
Line 34: "Prevotella species related ..." unsure about this statement as essentially the entire network of rumen microbes contribute to VFA synthesis in some way. Perhaps you were targeting specific VFA?
Introduction:
Line 45 and 46: Consider combining the two sentences. At the beginning of line 46 and extra "the" is included before food quality. Here and throughout there is some excess use of "the".
Line 51: This line is misleading as the rumen microbiome is responsible for much more than milk production. Later there is more development into those roles but this could perhaps be revised or removed.
Line 69: "microbiome" should be plural here?
Line 71 & 72: Although the reference may be specific to dairy cattle this capability is not limited to only dairy cattle but rather a result of the ruminant system in general.
Line 81: consider switching order of "yet not" to "not yet"
Line 82: This is more of a general comment but the lack of data at the species level could be a result of the uncertainty and lack of confidence in identifying microbes to that level considering databases that are ever growing and developing.
Line 83: word choice "constitutions", unsure if this is the proper use of the word here.
Materials and methods: In general there is some lack of detail in methods but can be readily revised.
Line 98: "the representative samples" is unclear.
Line 99: no information was provided on dissection techniques and/or landmarks.
Line 103: "conducted for 1 mL", conducted utilizing 1 mL?
Line 106-109: Perhaps the animal care statement should appear first in the methods.
Was there any mechanical lysis methods used before the DNA Stool Mini Kit? What was the starting amount of rumen fluid used for extraction?
Line 114: Is there an image of the gel available?
Line 114: What was the starting concentration of gDNA used for library prep? What was the library prep protocol followed?
Line 134: Were any other databased utilized for taxonomic assignment? SILVA, etc.?
Results:
Table 3: Is this a general VFA profile from a combined sample across all locations?
Line 176: "through" is perhaps is not the right word here?
Line 211-212: consider rephrasing to incorporate % more effectively with sentence flow.
Line 239: "ruminal microbiome" is perhaps the wrong word choice here but rather "microbes" or something to indicate specific taxa not the entire population.
Line 243: "Besides" in not perhaps a good transition word between thoughts.
Discussion: This section could utilize some general editing. Consider taking out "the" in main instances (Line 294, Lin 297, Line 309, etc). Also, predominant is utilized at many times but other synonymous word could be used in various places.
Line 306: Was Prevotella also the most abundant in the low milk protein group as well?
Line 326: What is meant by "higher advantage in genus content"?
Line 331: Microbes instead of microbiomes?
Line 332: abundance of "microbes" instead of microbiome?
Line 370-371: Why was rumen fluid not collected via oral lavage from many individuals?
Conclusions:
Line 383: Richness is often used more to describe alpha diversity differences rather than abundance differences.
General comment/questions: I am curious why the animals were sacrificed and why rumen fluid was collected from several locations in the rumen. Why were methods of oral lavage not considered? Were any controls utilize in this experiment in terms of sequencing controls (positive and negative control)?
Reviewer 2 Report
Dear Authors,
The manuscript titled “Identification of potential role of the rumen microbiome in modulating milk protein and fat in dairy cow using microgenomic sequencing” has been reviewed.
Thank you for this opportunity you have given me to review this manuscript, which successfully examined the relationship among the rumen microbiome in lactating Chinese Holstein cows with extremely high and low milk protein and fat percentages, ruminal fermentation, and milk quality parameters that might contribute to high-quality dairy milk production.
Generally, the manuscript is well presented starting from the introduction to the interesting gathered results and conclusion.
Minor comments:
Title
Line 3: “microgenomic”- better to use “metagenomic”. Metagenomic was often mentioned in the entire manuscript.
Abstract
Lines 25-26: Please reconstruct the sentence. VFAs are the contents of the rumen fluid and not the other way around.
Lines 28, 34: scientific names are in italic form
Line 29,30,34: Acronyms should be spelled-out in the first appearance in abstract, main text and tables/figures, and then abbreviation may use thereafter.
Introduction
Line 49: “Akin”. I suggest to use other word.
Lines 72,75, and elsewhere in the Discussion part: check the citation. Please revise according to the format.
Materials and Methods
Line 93: First, observe consistency in using terminologies. e.g. assigned acronym for your treatments. In lines 90-91, treatments were HPF and LPF denoting for high protein and fat, and low protein and fat, respectively; however, in Table 1, HFP and LFP were being used. These often see in the succeeding part of the manuscript. Please check very carefully.
In Table 1, use superscript to show significant difference between treatments especially to p < 0.05.
I also suggest to use 3 decimal places in your SEM and p values while 2 decimal places in treatment data. Please be consistent in all your tables.
Line 96: Please rephrase the sentence to prevent redundancy.
Line 141: Never use “Anosim analysis” because it is defined as Analysis of similarities. Use ANOSIM instead. Anosim analysis is often used elsewhere. Please check and change.
Results
Line 158: NH3-N should be NH3-N.
Line 160: Please check the treatments label (“HFP”, “LFP”). Refer to comments in Materials and Methods. Check all the details in the table.
Lines 168-170: Delete the statement since it is already mentioned in your materials and methods.
Line 176: supplementary figure s1 is not provided.
Lines 182-183: Delete. The same as with that of in lines 168-170.
Line 187: supplementary figure s2 not shown.
Lines 190-192: Scientific names should be in italic form. It is found elsewhere in the Results and Discussion part. Please italicize.
Line 204: Anosim analysis to ANOSIM
Line 209: isn’t it “R-value > 0 shows that inter-group differences are greater than intra-group differences.”
Lines 211-216: italicize
Line 216: HFP to HPF
Lines 224-225: italicize
Line 229: check the legend denoting treatments. Be consistent.
Line 234: italicize
Line 236: change the figure legend for treatments
Lines 240-243: italicize
Line 257: only 40 are shown in the figure. Please check.
Line 273: change the figure legend for treatments
Line 283: change the figure legend for treatments
Line 287: 5 AAs based on your data.
Line 292: change the figure legend for treatments
Discussion
Line 298: “6977 unique microorganisms”. Are you pertaining to observed OTUs here? If so, then better to use observed OTUs.
Lines 345-346: Please cite your reference.
Lines 348-349: Please cite your reference.
Lines 354-356: Please cite your reference.
Line 368: Is this figure lifted from other paper, if so, please cite the author. And if not, please specify in your discussion.
Lines 370: The authors properly mentioned the scope and limitation of the study.
Lines 376: Recommendation of the authors is properly presented.
Generally, please check and change to italic form all the scientific names.
Conclusion
This section is well presented. Just italicize the scientific name.
Reviewer 3 Report
The manuscript is well-written and the presented information is relevant for the field. However, there are several major weak points that prevent its publication in the present form.
First of all, I suggest the authors change the title of the article. Microgenomic sequencing doesn't exist such as specific sequencing, they should use metagenomic sequencing, metagenomics, whole metagenome sequencing, or similar. In addition, I think that they didn't obtain results about the "modulating" MP and FP, I suggest they think about the results obtained in this work and then making a great new title.
Moreover, half of the simple summary is incomplete for me, the authors said in line 18 that the abundance of Prevotella species in the rumen with high MP and FP what?? it seems to be missing a verb. In advance, I recommend didn't say that Prevotella species are related to the group HPF because the results showed that there are species of Prevotella also related to LPF (figure 3).
The authors should correct and explain first:
32: Spearman is a correlation analysis, please don't mistake correlation with an association, although the results may match between two analyses, they are not the same. Please clarify. Moreover, they didn't explain this analysis in the material and methods section.
37-38: Did you provide a basis that can improve the lactation performances in dairy cows? How? I think that this sentence is very risky to this work. Please change it.
46: Add a reference to the fact that MP and FP are the key factors determining milk quality. Milk quality might refer to somatic cell count. Please clarify.
80-82: Are you sure? Please check this review: https://doi.org/10.1080/19490976.2018.1505176
Bainbridge et al. related microbial species with different milk traits, and I think that further investigation is needed.
93: I think that a supplementary file that contains the raw data (complete for table1 or table2) for the 4000 cows might help readers.
95: Table2. Did you give more importance to PP or FP? The three most high cows match in PP and FP, because the order isn't the same. This is one of the reasons I before recommended authors add a supplementary file.
117: Please, remove PE150 in the name of the Illumina sequencing device.
122-125: what software did you use to remove host DNA? What aligner? Please add it.
137: Krona is a visualization software, hence, I suggest changing this sentence to, Krona was used to display....
139-140: Please add what taxonomical level did you use to carry out the PCA and NDMS plot.
142-143: The same as before, please specify the level to perform the LEfSe and Metastats analyses. I read that it is at the species level, however, the authors should indicate here.
168: You didn't sequence the microbiota DNA. When you sequence a rumen sample you sequence all DNA presented in the sample. Please clarify.
171: Check if the sequencing generates 76Gbp. The data provided in the supplementary file account 78Gbp.
171-175: it is not very clear which measurements the authors are referring to. And if this part is a material and method, please move to the correct section in the manuscript. It's very important this part I think, the authors should explain the correlation methods.
176: In the material and methods sections scaftigs were named, however in this part the authors didn't give the number of the total obtained scaftigs.
176: Supplementary figures aren't included in the manuscript or in the compress files.
184: please change kinds. It isn't kinds of...
215: Change kinds of microbiota.
219: Figure 3. The LDA SCORE (log 10) for de HPF group is represented in negative in this plot, however, in the supplementary file doesn't. Please clarify in the title of the supplementary table or change the values. The same for Figures 7,8, 9, and 10.
238: You referred to Spearman correlation analyses, nevertheless you didn't explain how did you do in the material and method section.
246: Figure 6: Please decrease the letter size of the plot to be more clear.
273: Figure 8. Please, check the legend of the plot, HFP or HPF? The same for figure 9 and 10.
324-325: The authors discuss the diversity of the two groups studied here, though the diversity doesn't appear in the results and eighter in the material and methods. They must include a supplementary file with this information.
For now, I can't make a greater review because I have to mention one of the most important problems of this manuscript. The authors assumed the limitation of the sample size of this study, however, I think that there is very limited analysis. On the one hand, I suggest that they should add references more actual that use and prove that use only 3 replicates is enough in this kind of data, shotgun metagenomics. On the other hand, I suggest the authors make an additional analysis for verifying the functional results (KEGG, eggNOG and CAZy). For the taxonomic data, the authors performed two analyses, LEfSe and Metastats, and they showed only the common results to get good approaches. Hence, for functional analyses, the authors need to add a second analysis (with DESeq or another differential abundance software) to confirm the results presented here.
380: Your conclusion should be more direct, clear, and not a discussion on the experiment and results. This is the conclusion
Round 2
Reviewer 3 Report
Thank you for making the recommended corrections in the manuscript and the requested analyses.
However, there are some relevant weak points. I will try to highlight them with the aim to help the authors to improve the quality of their presentation.
1) The authors added the requested analyses verifying the functional results reported here, thank you. Nevertheless, I need to know why are they inconsistent in de LDA filters. For example, you used the filter LDA >3 in L235, L298, L308 and LDA>2.0 in L273, L281. Could you explain why you use two different values based on the data?
2) L311-319. The authors claimed that some species are related to some functions (later, they will build their conclusions based on this). However, why did the authors assume that these functions are related to this microorganism? Only for the abundance? Are the revealed functions in the contigs of these species? I mean, have the authors confirm this statement?
3) Minor Points:
L46: the author changed the sentence and added a reference. However, I think that this reference isn't enough for explaining the sentence. You use a RNASeq paper to explain that these two traits are important, for me you need to change the reference to once matching better with the sentence (in global).
L133: Please remove "potential". They aren't the potential reads of the host, they are host reads. Simplify the sentence.
L157: the authors used ANOSIM, however, they don't describe the acronyms of the method around the manuscript. Please describe here.
L192: Are you assembling the sample, or the samples? In this part, it isn't clear how the authors did. Please clarify.
L202: When I said to the authors that should remove "kinds of" is for removing "kinds of" not only "of". It's a wrong expression of microorganisms. You can use, for example, once time but not only always in the manuscript.
L215: Please move the added part above because in lines 215-217, it breaks the paragraph in which the author explains the PCA.
L227: Anosim is lower case. Please be consistent.
L410: Microgenomics don't exist. Please remove or change it.
